# A Magnetic Field Containment Method for an IPT System with Multiple Transmitting Coils Based on Reflective Properties

Xu Yang [1,2], Junfeng Yang [1,*], Jing Fan [3], Bao Wang [3] and Dingzhen Li [3]

1    School of Electrical Engineering, Beijing Jiaotong University, Beijing 100044, China
2    School of Intelligent Manufacturing, Nanyang Institute of Technology, Nanyang 473004, China
3    School of Information Engineering, Nanyang Institute of Technology, Nanyang 473004, China
*    Correspondence: yangjunfeng@bjtu.edu.cn

**Abstract:** Inductive power transfer (IPT) systems with multiple transmitting coils are mainly used in specific scenarios, such as IPT sharing platforms and dynamic wireless charging of electric vehicles, etc. However, it faces problems of electromagnetic field leakage and low efficiency. A new magnetic field containment method based on reflective properties is proposed to solve the above shortcomings. Firstly, the reflective properties and performance figures of the IPT system with a unified passive compensation network are described and derived. Then, an S−LCL topology appropriate for the time−varying coupling IPT system is presented, where the IPT system's transmitter consists of multiple coils that are compatible with one or more moving receivers and is powered by an inverter. Then, magnetic field focusing, power transfer and overall efficiency are analyzed and simulated. Finally, an experimental prototype is built to validate the feasibility of the proposed system. The experimental results show that the proposed method can increase the power transfer of the coupled transmitting coil and reduce the magnetic field leakage of the standby transmitting coils without complex shielding measures, switch, position detection and communication circuits.

**Keywords:** inductive power transfer; self−regulating magnetic field; reflective properties; multiple transmitting coils; S−LCL topology

## 1. Introduction

Inductive power transfer (IPT) technology is considered to be a good solution to replace the traditional electrical connection method due to its advantages of security, flexibility, aesthetics and reliability [1]. With the development of technology and the expansion of the application field, an IPT system with multiple transmitting coils has become a research hotspot [2–4]. It is mainly used in specific scenarios, such as IPT sharing platforms for electronic devices, dynamic wireless charging for electric vehicles or automated guided vehicles, etc. [5,6]. The IPT sharing platform for mobile phones is shown in Figure 1. It includes multiple transmitting coil arrays which can charge one or more phones at the same time [7]. The concept diagram of a dynamic IPT system is shown in Figure 2, and the transmitting coil (TC) is selectively coupled to the moving receiving coil (RC) as the car passes through the charging section. The common point of the two application scenarios is that the transmitter of the IPT system includes any number of transmitting coils according to the application needs, and the receiver includes one or more receiving coils as the space permits. Compared with an IPT system with one transmitting coil, the IPT system with multiple transmitting coils has experienced some new technical problems, such as complex coordination and control among the transmitting coils, electromagnetic field (EMF) leakage of the standby transmitting coils, low system efficiency, etc. [8].

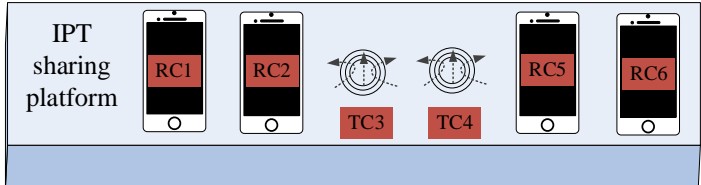

**Figure 1.** IPT sharing platform for mobile phones.

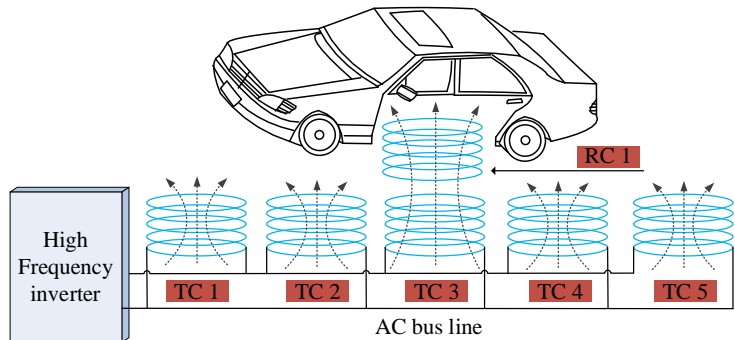

**Figure 2.** Dynamic wireless charging for electric vehicles.

IPT systems with multiple transmitting coils can be roughly classified into three categories according to the arrangement of transmitting coils. First, a long transmitting coil is used to replace multiple transmitting coils [9,10]. The single−coil design simplifies control complexity and provides a relatively constant coupling coefficient for the receiving coil. However, EMF leakage may interfere with nearby electronic equipment. Second, the IPT system with multiple transmitting coils and multiple inverters is proposed [11,12]. However, this scheme is complicated because each transmitting coil requires a separate inverter, a position detection circuit, and a switching element. Third, multiple transmitting coils are connected to a single inverter in parallel, which effectively reduces the number of inverters (as shown in Figure 2). However, auxiliary circuits, such as position detection and switches, are still essential [13,14].

In [15,16], a magnetic field control method was proposed to realize efficient power transfer without switches and control circuits by utilizing the reflective properties of the IPT system. However, only the receiver with two capacitors for the series and parallel compensation was discussed. The reflective properties of the receiver with other topologies were analyzed in [17,18]. However, the transmitter was not compensated in any way. An auto−tuning control system by variation of self−inductance for dynamic wireless electric vehicle charging was proposed in [19], which also explored the application of reflective properties in the IPT system with multiple transmitting coils, but the primary and secondary coils needed to contain magnetic materials. An IPT system with dual secondary loops which could automatically control the current of the transmitting coil by its reflective property was presented in [20], but a capacitor array was added in the receiver, which increased the complexity of the system.

In summary, the segmentation (magnetic field containment) methods for IPT systems with multi−transmitting coils can be classified into two groups: a segmentation method based on position detection and switch control, and an adaptive segmentation method based on reflective properties (automatic segmentation according to the position of the transmitting coil). The adaptive segmentation method can simplify the control strategy and improve the reliability and stability of the IPT system, but it has problems such as a limited application range of load, the need for additional magnetic materials or capacitor arrays. Therefore, it is necessary to continue to study the new adaptive segmentation−type compensation topology and its design method parameters.

The primary side of S−LCL compensation topology is simple and the secondary side is convenient to adjust. With reasonable parameter design, the S−LCL compensation

topology can be used in load−independent constant voltage output [21,22], output voltage identification [23] and other application scenarios. To the best of our knowledge, the S−LCL compensation topology is first applied to the IPT system with multi−transmitting coils to realize adaptive segmentation. When the receiving coil approaches the transmitting coil, the transmitting coil current automatically increases. Conversely, when the receiving coil moves away from the transmitting coil, the transmitting coil current automatically decreases. The proposed methods do not require additional switches and position detection circuits compared to traditional methods, so the control strategy is simplified and the reliability is improved. In addition, EMF leakage is effectively reduced, and the overall efficiency of the system is also improved.

The rest of this paper is organized as follows. Section 2 provides the reflective properties description and performance figures of the unified passive compensation network. The S−LCL topology and the parameter design method are presented in Section 3. The system's design and analysis are described in Section 4. Experimental verification and further discussion are presented in Section 5. Finally, Section 6 draws the conclusions of this paper.

## 2. Reflective Segmentation Basic

### 2.1. Equivalent Circuit of the IPT System with Unified Passive Compensation Network

The schematic diagram of the IPT system is shown in Figure 3, which is mainly composed of a DC voltage source, a full bridge inverter, primary and secondary compensation networks, transmitting and receiving coils, a rectifier, a filter and a load. Here, $U_D$ is the DC input voltage. $Q_1$–$Q_4$ are four switches, which form a full bridge inverter; $u_{AB}$ and $i_{AB}$ represent its instantaneous output voltage and current, respectively. The compensation networks are composed of one or more inductors or capacitors. $L_P$, $R_P$ are the inductance and parasitic resistance of the transmitting coil, respectively. $L_S$ and $R_S$ are the inductance and parasitic resistance of the receiving coil, respectively. $M$ represents mutual inductance between the coils, which is described as $M = k\sqrt{L_P L_S}$, where $k$ is the coupling coefficient. $D_1$–$D_4$ are four diodes that constitute a rectifier; $u_{ab}$ and $i_{ab}$ represent its instantaneous input voltage and current, respectively. The capacitance $C_F$ is the filter, and $R_L$ is the load.

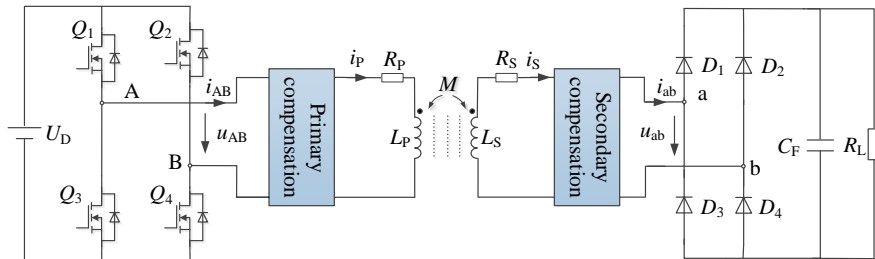

**Figure 3.** Schematic diagram of the IPT system with a unified passive compensation network.

The fundamental harmonic approximation method is adopted in this paper. Here, $U_{AB}$, $I_{AB}$, $U_{ab}$, and $I_{ab}$ are, respectively, the root mean square (RMS) values of the fundamental harmonics of $u_{AB}$, $i_{AB}$, $u_{ab}$, and $i_{ab}$. $I_P$ and $I_S$ are the RMS values of the current flowing through the transmitting and receiving coils, respectively. Then, the output voltage $U_{AB}$ of the inverter can be calculated by Equation (1).

$$U_{AB} = \frac{2\sqrt{2}}{\pi}U_D \tag{1}$$

According to the equivalent conversion relationship of rectifier and filter, the input voltage $U_{ab}$ and current $I_{ab}$ of the rectifier can be obtained [15].

$$\begin{cases} U_{ab} = \frac{2\sqrt{2}}{\pi}U_{RL} \\ I_{ab} = \frac{\pi\sqrt{2}}{4}I_{RL} \end{cases} \tag{2}$$

The rectifier bridge and the resistance load $R_L$ are equivalent to the load $R_{eq}$ of the secondary side, which can be expressed by Equation (3).

$$R_{eq} = \frac{8}{\pi^2} R_L \tag{3}$$

According to the reciprocity theorem, any passive linear network can be modeled with a T−type or a π− type network. Therefore, a T−type circuit is selected to describe the passive compensation network in this paper. Based on Equations (1)–(3), the equivalent circuit of the IPT system with a unified passive compensation network can be obtained, which is shown in Figure 4. $Z_{P1}$, $Z_{P2}$ and $Z_{P3}$ are the impedance of a single compensation inductor or capacitor in the primary compensation network. $Z_{S1}$, $Z_{S2}$ and $Z_{S3}$ are the impedance of a single compensation inductor or capacitor in the secondary compensation network. Here, $Z_{pi} = jX_{Pi} + R_{Pi}$, $Z_{Si} = jX_{Si} + R_{Si}$, (i = 1,2,3), where $X_{Pi}$ and $X_{Si}$ represent the reactance of the compensation inductor or capacitor, $R_{Pi}$ and $R_{Si}$ represent the parasitic equivalent series resistance (ESR) of the compensation inductor or capacitor.

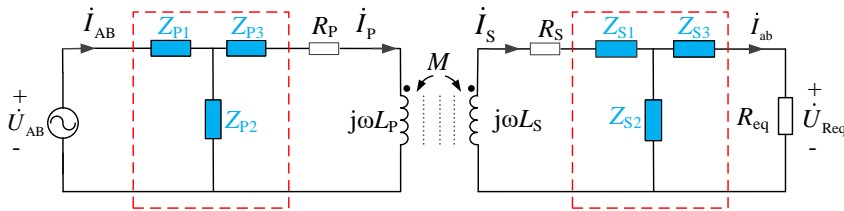

**Figure 4.** Equivalent circuit of the IPT system with a unified passive compensation network.

*2.2. Reflective Properties and Performance Figures*

The impedance model of the IPT system is established according to mutual inductance theory, which can be calculated by Equation (4).

$$\begin{cases} Z_P = \frac{Z_{P2}(Z_{P3}+j\omega L_P+R_P)}{Z_{P2}+Z_{P3}+j\omega L_P+R_P} + Z_{P1} \\[2mm] Z_S = \frac{Z_{S2}(Z_{S3}+R_{eq})}{Z_{S2}+Z_{S3}+R_{eq}} + j\omega L_S + R_S + Z_{S1} \\[2mm] Z_r = \frac{(\omega M)^2}{Z_S} \\[2mm] Z_{in} = \frac{Z_{P2}(Z_{P3}+j\omega L_P+R_P+Z_r)}{Z_{P2}+Z_{P3}+j\omega L_P+R_P+Z_r} + Z_{P1} \end{cases} \tag{4}$$

where $Z_P$ is the impedance of the primary side when $k = 0$. $Z_S$, $Z_r$ and $Z_{in}$ represent the secondary side impedance, reflected impedance and input impedance, respectively. $\omega = 2\pi f$ is the angular frequency of the power source, where $f$ represents the frequency. The currents for each branch can be obtained, as shown in Equation (5).

$$\begin{cases} \dot{I}_{AB} = \frac{\dot{U}_{AB}}{Z_{in}} \\[2mm] \dot{I}_P = \frac{Z_{P2}\dot{I}_{AB}}{Z_{P2}+Z_{P3}+j\omega L_P+R_P+Z_r} \\[2mm] \dot{I}_S = \frac{\dot{U}_S}{Z_S} = \frac{j\omega M \dot{I}_P}{Z_S} \\[2mm] \dot{I}_{ab} = \frac{Z_{S2}\dot{I}_S}{Z_{S2}+Z_{S3}+R_{eq}} \end{cases} \tag{5}$$

Similarly, the output voltage can be obtained, which is shown in Equation (6).

$$\dot{U}_{ab} = \frac{j\omega M Z_{S2} Z_{P2} \dot{U}_{AB}}{Z_{in}Z_S(Z_{S2}+Z_{S3}+R_{eq})(Z_{P2}+Z_{P3}+j\omega L_P+R_P+Z_r)}R_{eq} \tag{6}$$

when $k = 0$, the current flowing through the transmitting coil is

$$\dot{I}_{P,un} = \frac{\dot{U}_{AB}}{Z_P} = \frac{\dot{U}_{AB}(Z_{P2} + Z_{P3} + j\omega L_P + R_P)}{Z_{P1}(Z_{P2} + Z_{P3} + j\omega L_P + R_P) + Z_{P2}(Z_{P3} + j\omega L_P + R_P)} \tag{7}$$

The constant $k_0$ is defined as the maximum coupling coefficient of the system. $Z_{r,k}$ is the reflective impedance when the coupling coefficient is $k$. $Z_{in,k}$ is the input impedance when the coupling coefficient is $k$. When $0 < k \le k_0$, the current flowing through the transmitting coil is

$$\dot{I}_{P,co} = \frac{\dot{U}_{AB}}{Z_{in,k}} = \frac{\dot{U}_{AB}(Z_{P2} + Z_{P3} + j\omega L_P + R_P + Z_{r,k})}{Z_{p1}(Z_{P2} + Z_{P3} + j\omega L_P + R_P + Z_{r,k}) + Z_{P2}(Z_{P3} + j\omega L_P + R_P + Z_{r,k})} \tag{8}$$

As can be seen from Equations (4), (7) and (8), the transmitting coil current is related to the reflected impedance, which is related to coupling coefficient $k$. According to Ampere's law, the transmitting coil current is related to the EMF leakage. Therefore, the IPT system should adjust the transmitting coil current to satisfy EMF standards. In order to further describe the reflective properties, the segmentation ratio (SR) is defined as the ratio of the coupled branch current to the uncoupled branch current, which can be described as

$$SR \triangleq \frac{\left|\dot{I}_{P,k0}\right|}{\left|\dot{I}_{P,un}\right|} = \frac{Z_{in,un}}{Z_{in,k0}} = \frac{Z_P}{Re[Z_{r,k0}]} \tag{9}$$

where $\dot{I}_{P,k0}$ is transmitting coil current when $k = k_0$. $\dot{I}_{P,un}$ is the transmitting coil current when $k = 0$. When the voltage $U_{AB}$ is constant, the transmitting coil current can be adjusted by the input impedance. The parameter configuration method designed in this paper is as follows. When $k = 0$, the input impedance of the system is $Z_{in,un}$, i.e., $Z_P$, which is inductive and the maximum. When $k = k_0$, the input impedance of the system is $Z_{in,k0}$, which satisfies the relation $Z_{in,k0} = Re[Z_{r,k0}]$. Here, $Re[Z_{r,k0}]$ represents the real part of the reflected impedance $Z_{r,k0}$. When $0 < k < k_0$, the input impedance of the system is $Z_{in,co}$, which is closely related to the coupling coefficient $k$.

Through reasonable topology design and the parameter configuration method, the transmitting coil current can be automatically adjusted by the position of the receiving coil. This means that the current of the working coil and the standby coil can be automatically segmented without additional complicated shielding measures, a switch, position detection and communication circuits. The detailed topology design and parameters configuration method are described in Section 3.

## 3. Proposed Topology and Operation

The primary side of the S−LCL compensation topology is simple, and the secondary side is convenient to adjust [21]. Through proper configuration, no−load overcurrent can be avoided. Therefore, the S−LCL topology is adopted in this paper. As shown in Figure 5, the IPT system consists of a transmitter and a receiver, the transmitter is composed of multiple transmitting coils which are connected in parallel to a single inverter, and the receiver is composed of a receiving coil, a compensation element and a load, which may be one or more. In order to simplify the analysis, assuming that each transmitting coil is identical to the others, the mutual inductance between the adjacent transmitting coils can be ignored, and the parasitic parameter of each component is ignored.

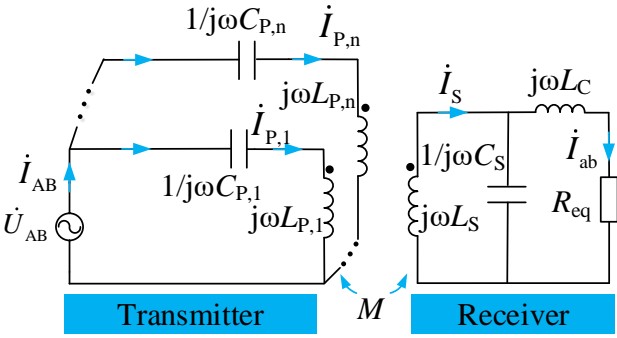

**Figure 5.** Equivalent circuits of the proposed IPT system.

*3.1. Receiver Design*

The receiver in Figure 5 uses the LCL topology. A two−port network model is established, which can be obtained from Equation (10).

$$
\begin{bmatrix} 1 & 0 & 0 & 0 \\ 1 & 0 & -Z_{11} & -Z_{12} \\ 0 & 1 & -Z_{21} & -Z_{22} \\ 0 & 1 & 0 & R_{eq} \end{bmatrix} \begin{bmatrix} \dot{U}_S \\ \dot{U}_{ab} \\ \dot{I}_S \\ \dot{I}_{ab} \end{bmatrix} = \begin{bmatrix} \dot{U}_S \\ 0 \\ 0 \\ 0 \end{bmatrix} \tag{10}
$$

where $Z_{11}$, $Z_{12}$, $Z_{21}$ and $Z_{22}$ are the Z parameters of the two−port, and $\dot{U}_S$ is the induced voltage of the secondary side, i.e., $\dot{U}_S = j\omega M \dot{I}_P$. The impedance $Z_S$, the voltage $\dot{U}_{ab}$ and the current $\dot{I}_{ab}$ can be obtained, as shown in Equation (11).

$$
\begin{cases} Z_S = \dfrac{R_{eq} + j\omega(L_C A - C_S R_{eq}^2 + L_S A^2 + L_S B^2)}{A^2 + B^2} \\[2mm] \dot{I}_{ab} = \dfrac{\dot{U}_S}{R_{eq}(1 - L_S C_S \omega^2) - j(L_S C_S L_C \omega^3 - L_S \omega - L_C \omega)} \\[2mm] \dot{U}_{ab} = \dfrac{R_{eq} \dot{U}_S}{R_{eq}(1 - L_S C_S \omega^2) - j(L_S C_S L_C \omega^3 - L_S \omega - L_C \omega)} \end{cases} \tag{11}
$$

where $A = 1 - L_C C_S \omega^2$, $B = R_{eq} C_S \omega$. According to Equation (11), when the real part of the denominator for $I_{ab}$ is zero, the output current is independent of the load. The output current and frequency can be obtained, as shown in Equation (12).

$$
\begin{cases} \dot{I}_{ab} = \dfrac{\dot{U}_S}{j\omega L_S} \\[2mm] \omega = 2\pi f = \dfrac{1}{\sqrt{L_S C_S}} \end{cases} \tag{12}
$$

Define the constant $\alpha = L_S/L_C$. According to Equation (11), when the imaginary part of the denominator for $\dot{U}_{ab}$ is zero, the output voltage is independent of the load. The output voltage and resonant angular frequency are given as Equation (13).

$$
\begin{cases} \dot{U}_{ab} = -\dfrac{\dot{U}_S}{\alpha} \\[2mm] \omega = 2\pi f = \sqrt{\dfrac{L_S + L_C}{L_S C_S L_C}} = \sqrt{\dfrac{1 + \alpha}{L_S C_S}} \end{cases} \tag{13}
$$

In this case, the secondary side impedance $Z_S$ of the LCL compensation topology can be written as Equation (14).

$$
Z_S = \frac{R_{eq} + j\omega\alpha C_S R_{eq}^2}{(1/\alpha)^2 + (1 + \alpha)(C_S/L_S)R_{eq}^2} \tag{14}
$$

Based on Equations (4) and (14), the reflected impedance can be obtained, which can be written as Equation (15).

$$Z_r = \frac{(\omega M)^2}{\alpha^2 R_{eq}} - j\frac{1}{\alpha} C_S M^2 \omega^3 = \frac{(\omega M)^2}{\alpha}(\frac{1}{\alpha R_{eq}} - jC_S\omega) = R_r - jX_r \tag{15}$$

If the induced voltage $\dot{U}_S$ of the secondary side of the IPT system is unchanged, the constant voltage output can be realized according to Equation (13). This paper studies the IPT system with variable coupling coefficients, so it focuses on obtaining the reflective impedance, which is required by the IPT system through proper parameter configuration of $C_S$, $L_S$ and $L_C$.

*3.2. Transmitter Design*

As can be seen from Figure 5, the transmitter is composed of a series of transmitting branches, each of which consists of a transmitting coil and a compensation network, and an arbitrary number of transmitting branches can be connected in parallel according to requirements.

When the receiver is decoupled from the transmitting branch, the input impedance is the maximum and inductive, which can be described as follows:

$$Z_{in,un} = j\omega L_{P1} + \frac{1}{j\omega C_{P1}} = X_{r,k0} = j\frac{1}{\alpha} C_S M_0^2 \omega^3 \tag{16}$$

where $k_0$ and $M_0$ represents the coupling coefficient and mutual inductance in the condition of perfect alignment, respectively, and satisfies the relation $M_0 = k_0\sqrt{L_P L_S}$. $X_{r,k0}$ is the reflected reactance when $k = k_0$.

$$Z_{in,co} = j\omega L_{P1} + \frac{1}{j\omega C_{P1}} + Z_r \tag{17}$$

when the branch of the transmitter is fully coupled with a receiving coil, the impedance of the transmitter branch is pure resistance, which can be expressed by Equation (18).

$$Z_{in,k0} = j\omega L_{P1} + \frac{1}{j\omega C_{P1}} + Z_{r,k0} = R_{r,k0} = \frac{(\omega M_0)^2}{\alpha^2 R_{eq}} \tag{18}$$

where

$$Z_{r,k0} = \frac{(\omega M_0)^2}{\alpha^2 R_{eq}} - j\frac{1}{\alpha} C_S M_0^2 \omega^3 = R_{r,k0} - jX_{r,k0} \tag{19}$$

Therefore, the value of the compensation capacitor can be obtained from Equation (20).

$$C_{P1} = \frac{1}{\omega^2(L_{P1} - \frac{1}{\alpha} C_S M_0^2 \omega^2)} \tag{20}$$

If the voltage $U_{AB}$ is constant, the transmitting coil current depends on the input impedance. According to the definition of *SR* in Equation (9) and impedance values in Equations (16) and (18), the *SR* of the IPT system can be calculated as Equation (21).

$$SR = \frac{\left|\dot{I}_{P,kmax}\right|}{\left|\dot{I}_{P,un}\right|} = \frac{\left|\dot{U}_{AB}/R_{r,kmax}\right|}{\left|\dot{U}_{AB}/X_{r,kmax}\right|} = \frac{X_{r,kmax}}{R_{r,kmax}} = \alpha(1+\alpha)\frac{1}{Q} \tag{21}$$

where $Q$ is the quality factor of the LCL compensated receiver, i.e., $Q = \omega L_S/R_{eq}$. Therefore, the *SR* can be adjusted by $\alpha$ and $Q$ according to actual need.

## 4. System Design and Analysis

### 4.1. Transmitter Design

By Ampere's law, the current of the transmitting coil is positively correlated with the EMF leakage. In order to describe the magnetic field focusing ability of the proposed system, the working characteristics are analyzed and compared with the commonly used LCC−P topology and S−P topology. The circuit schematics of two topologies are shown in Figure 6.

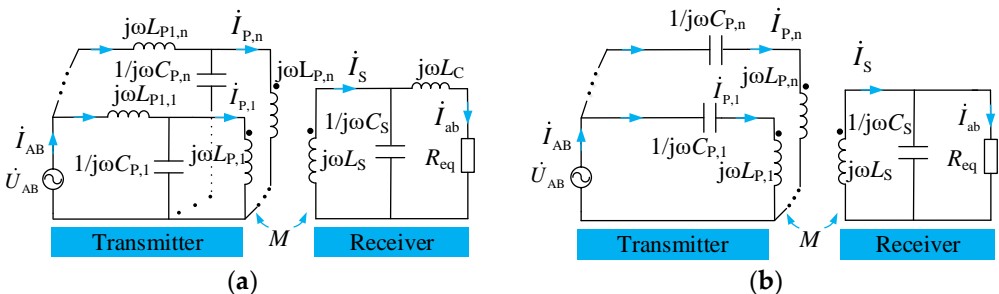

**Figure 6.** The commonly used compensation methods: (**a**) LCC−P topology; (**b**) S−P topology.

Where the parameters tuning method of the LCC−P topology can be obtained by Equation (22).

$$
\begin{cases}
C_{P1} = \dfrac{1}{\omega^2 L_{P1}} \\[2mm]
C_{P2} = \dfrac{1}{\omega^2 (L_P - L_{P1})} \\[2mm]
C_S = \dfrac{1}{\omega^2 L_S}
\end{cases}
\tag{22}
$$

Similarly, the parameters tuning method of the S−P topology can be obtained by Equation (23).

$$
\omega = \frac{1}{\sqrt{L_P C_P}} = \frac{1}{\sqrt{L_S C_S}}
\tag{23}
$$

The system parameters of the three topologies are the same; that is, $L_P = L_S = 270\ \mu H$, $k_0 = 0.3$, $f = 85\ kHz$. The IPT systems of the three topologies are nominally designed to transfer about 60 W to the load, and the compensation parameters are shown in Table 1.

**Table 1.** Parameters related to the three topologies.

| Parameters | LCC−P Topology | S−P Topology | S−LCL Topology |
|---|---|---|---|
| $U_D$ | 50 V | 22.5 V | 22.5 V |
| $L_{P1}$ | 72 μH | − − − | − − − |
| $C_{P1}$ | 48.71 nF | 14.27 nF | 14.76 nF |
| $C_{P2}$ | 17.71 nF | − − − | − − − |
| $C_S$ | 12.99 nF | 12.99 nF | 51.95 nF |
| $L_C$ | − − − | − − − | 90 μH |
| $R_{eq}$ | 200 Ω | 200 Ω | 36 Ω |

As shown in Table 1, the S−LCL topology satisfies the relations $\alpha = L_S/L_C = 3$ and $Q = \omega L_S/R_{eq}$. The *SR* of the three systems can be obtained by drawing the current curve variation with a frequency at $k = 0$, $k = k_0/2$ and $k = k_0$. As shown in Figure 7a, the transmitting coil current with the LCC−P topology is constant at the operating point of 85 kHz, which means that the *SR* is equal to 1. As can be seen from Figure 7b, the transmitting coil current with the S−P topology in the uncoupled state is the smallest at the operating frequency, which means that the *SR* is less than 1. For the S−LCL topology, the maximum current of the transmitting coil in the uncoupled state is shifted further from 85 kHz, resulting in a lower current of the standby transmitting coils (as shown

in Figure 7c). Therefore, the proposed S−LCL topology achieves the purpose of "field focusing" compared with the LCC−P and S−P topologies.

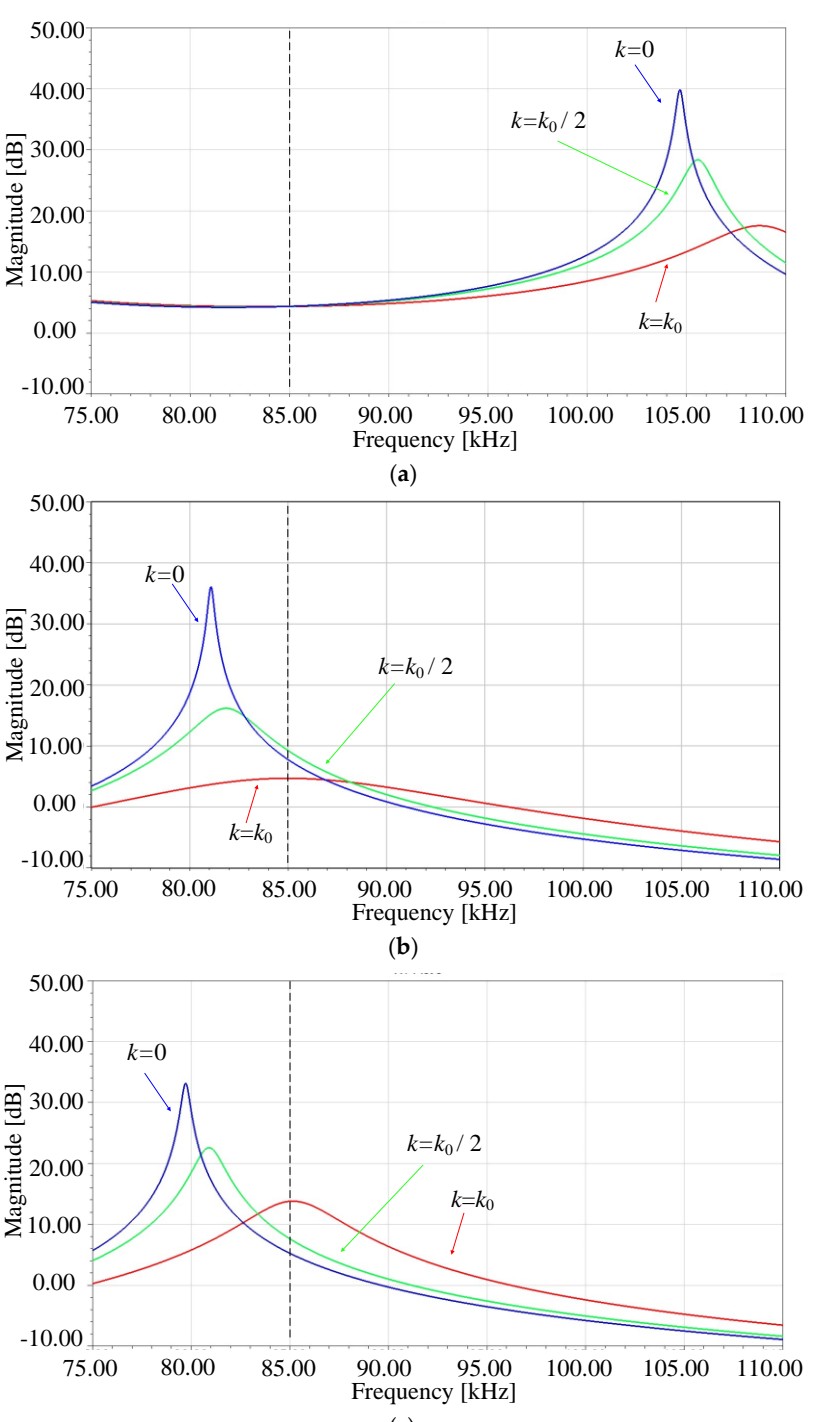

**Figure 7.** Transmitter coil current as a function of *k* and *f*. (**a**) LCC−P topology; (**b**) S−P topology; (**c**) S−LCL topology.

For comparison, the *SR* of the S−P and S−LCL topologies can be written together, as shown in Equation (24).

$$\begin{cases} \mathrm{SR_{S-P}} = \dfrac{I_{\mathrm{P,k0}}}{I_{\mathrm{P,un}}} = \dfrac{X_{\mathrm{r,k0}}}{R_{\mathrm{r,k0}}} = \dfrac{1}{Q_{\mathrm{P}}} \\[2mm] \mathrm{SR_{S-LCL}} = \dfrac{I_{\mathrm{P,k0}}}{I_{\mathrm{P,un}}} = \dfrac{X_{\mathrm{r,k0}}}{R_{\mathrm{r,k0}}} = \alpha(1+\alpha)\dfrac{1}{Q} \end{cases} \tag{24}$$

where $Q_P$ is the quality factor S−P topology, which is defined as $Q_P = R_{eq}/\omega L_S$. The S−LCL topology provides an extra degree of freedom over the S−P topology. By choosing appropriate $\alpha$ and $Q$, the desired $SR$ can be obtained, as shown in Figure 8.

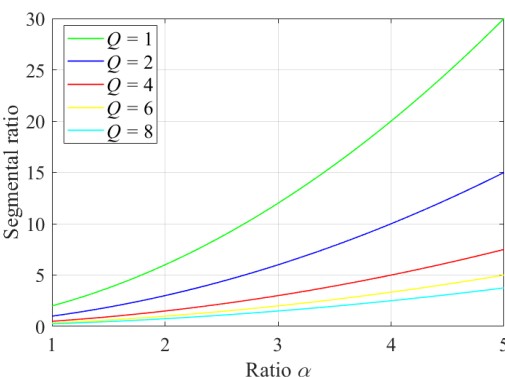

**Figure 8.** Effect of $\alpha$ and Q on the segmentation ratios.

### 4.2. Power Transfer and Efficiency Considerations

The power transfer of the three topologies, which is a function of the transmitting coil current and the real part of the reflected impedance, should also be discussed. In order to simplify the analysis, the parasitic resistances of the coils and the compensating elements are ignored. For the IPT system with LCC−P topology, the transmitting coil current and power transfer can be obtained from Equation (25).

$$\begin{cases} \dot{I}_P = \dfrac{\dot{U}_{in}}{j\omega L_{P1}} \\ P = (I_P)^2 k^2 \omega_0 L_P Q_P \end{cases} \tag{25}$$

For the IPT system with S−P topology, the transmitting coil current and power transfer can be obtained from Equation (26).

$$\begin{cases} \dot{I}_P = \dfrac{\dot{U}_{in}}{\omega_0 L_P \left[ k^2 Q_P + j(k_0^2 - k^2) \right]} \\ P = \dfrac{U_{in}^2}{(k^2 Q_P)^2 + (k_0^2 - k^2)^2} \dfrac{k^2 Q_P}{\omega_0 L_P} \end{cases} \tag{26}$$

Similarly, the transmitting coil current and power transfer for the S−LCL topology can be obtained from Equation (27).

$$\begin{cases} \dot{I}_P = \dfrac{\dot{U}_{in}}{\left[ k^2 Q + j\alpha(1+\alpha)(k_0^2 - k^2) \right]} \dfrac{\alpha^2}{\omega_0 L_P} \\ P = \dfrac{U_{in}^2}{(k^2 Q)^2 + \left[ \alpha(1+\alpha)(k_0^2 - k^2) \right]^2} \dfrac{\alpha^2 k^2 Q}{\omega_0 L_P} \end{cases} \tag{27}$$

Based on Equations (25)–(27), the relation curves of the transmitting coil current and power with the coupling coefficient can be obtained, as shown in Figures 9 and 10, respectively. As shown in Figure 9, when the coupling coefficient varies, the transmitting coil current for the LCC−P topology remains unchanged. The transmitting coil current for the S−P topology is the minimum when the coupling coefficient is maximum, indicating that the S−P topology is not conducive to transfer power and to prevent EMF leakage. For the S−LCL topology, the transmitting coil current decreases with the decrease in the coupling coefficient, thus limiting the EMF leakage and reducing overall losses.

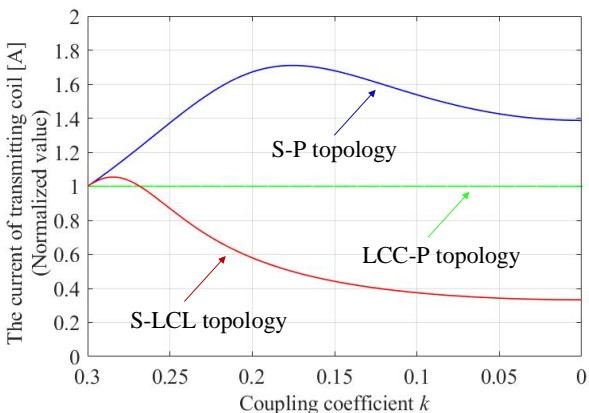

**Figure 9.** Effect of k on the current of transmitter coil.

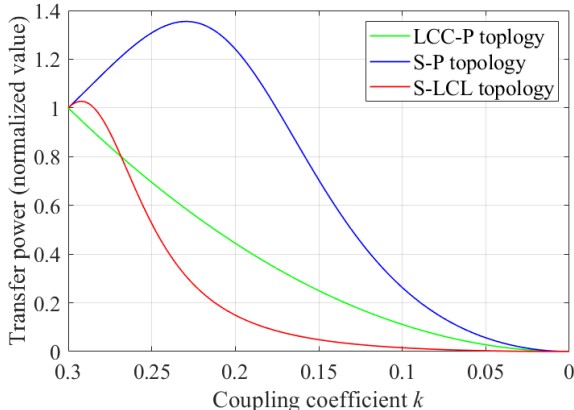

**Figure 10.** Effect of coupling coefficient *k* on the power transfer.

Since the current of the transmitting coil with the LCC−P topology is basically unchanged with the coupling coefficient, the "idle" losses are large. The current of the transmitting coil for the S−P topology with no load is usually larger than that with load, resulting in greater "idle" losses. Therefore, when the coupling coefficient is small, most of the power transfer for the LCC−P and S−P topologies becomes losses in the form of heat energy. The S−LCL topology can transfer high power when the coupling coefficient is large, and the output power sharply decreases with the small coupling coefficient. As a result, the losses of the standby transmitter coils are reduced and overall efficiency is improved. It should be noted that the curve of power transfer for the S−LCL topology can be appropriately adjusted by $\alpha$ and $Q$, and the high−power transfer area can be widened or narrowed by the coil design, which is beyond the scope of this paper.

## 5. Experimental Validation

In order to verify the correctness of the theoretical analysis, an experimental prototype was built, which is shown in Figure 11. It consisted of IT6523D DC power supply, a driver, a full−bridge inverter, transmitting and receiving coils, a compensation network, a diode rectifier, a M9715B DC electronic load, a Tektronix MDO3052 oscilloscope, a HIOKI PW6001 power analyzer, etc. Here, the MOSFETs of the inverter were C3M0120090; the rectifier diodes were selected as IDW20G65C5SiC. The planar circular coil was selected as the magnetic coupling mechanism because of its wide application.

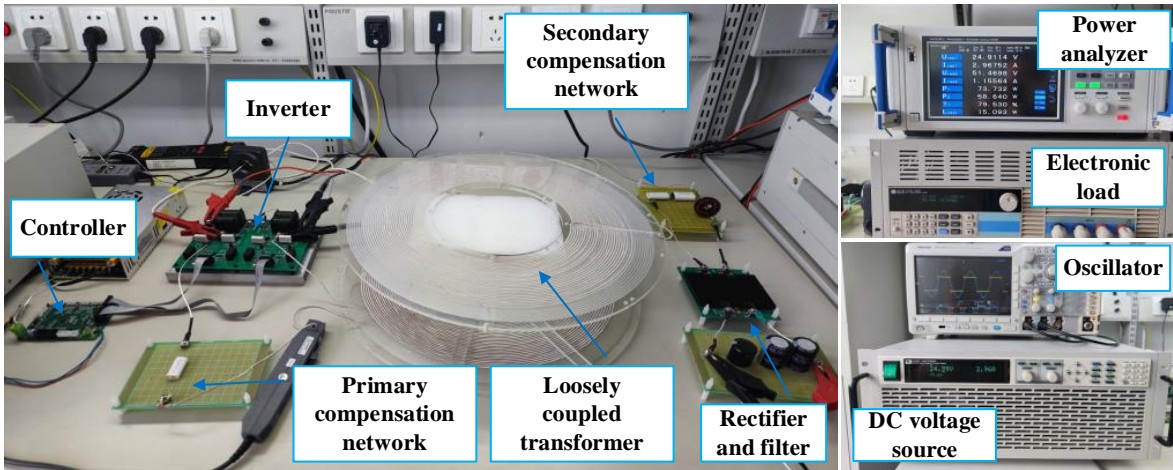

**Figure 11.** The realized prototype of the IPT system.

Based on the assumption in Section 3, the distance between the adjacent transmitting coils is large, and the mutual inductance can be neglected. Therefore, it is feasible to use a single transmitting coil and a single receiving coil in this experiment to verify the theoretical analysis. The parameters of the proposed system are $U_D = 25$ V, $L_P = L_S = 270$ μH, $k_0 = 0.3$ and $f = 85$ kHz, respectively. The output power is different under different $\alpha$ and Q, and the maximum output power of the prototype is about 100 W. The other parameters of the experimental prototype are listed in Table 2.

**Table 2.** Parameters related to the proposed system.

| Parameters | $\alpha = 2, Q = 2$ | $\alpha = 3, Q = 2$ | $\alpha = 3, Q = 4$ |
|---|---|---|---|
| $C_P$ | 15.02 nF | 14.76 nF | 14.76 nF |
| $C_S$ | 38.96 nF | 51.95 nF | 51.95 nF |
| $L_C$ | 135 μH | 90 μH | 90 μH |
| $R_{eq}$ | 72 Ω | 72 Ω | 36 Ω |
| $I_{P,k0}$ | 3.10 A | 6.69 A | 3.45 A |
| $I_{P,un}$ | 1.01 A | 1.12 A | 1.12 A |
| $SR$ | 3.07 | 5.97 | 3.08 |
| $P_{in}$ | 68.4 W | 135.6 W | 80.2 W |
| $P_{out}$ | 57.9 W | 101.8 W | 66.2 W |
| $P_{loss}$ | 10.2 W | 33.8 W | 13.9 W |
| $\eta$ | 84.6% | 75.1% | 82.6% |
| $P_{idle-loss}$ | 0.69 W | 0.85 W | 0.85 W |

When $\alpha = 3$, the measured waveforms and results are shown in Figure 12. The voltage and current waveforms of the inverter are measured by the oscillator and the DC−DC efficiencies are measured by the power analyzer. Among the measured items, $U_{rms1}$, $I_{rms1}$ are input DC voltage and current, $U_{rms2}$, $I_{rms2}$ are output DC voltage and current. $P_1$ and $P_2$ represent input and output power, respectively; $\eta_1$ and $L_{oss1}$ represent the efficiency and losses of the system. As shown in Figure 12a,b, the phase angle between the voltage and current waveforms of the inverter is almost zero. It indicates that the input impedance of the system is pure resistance in the fully coupled state. When the system works in the uncoupled state, the phase angle between the voltage and current waveforms of the inverter is almost 900 (as shown in Figure 12c), indicating that the input impedance of the system is inductance, which is consistent with the theoretical analysis in Section 3.

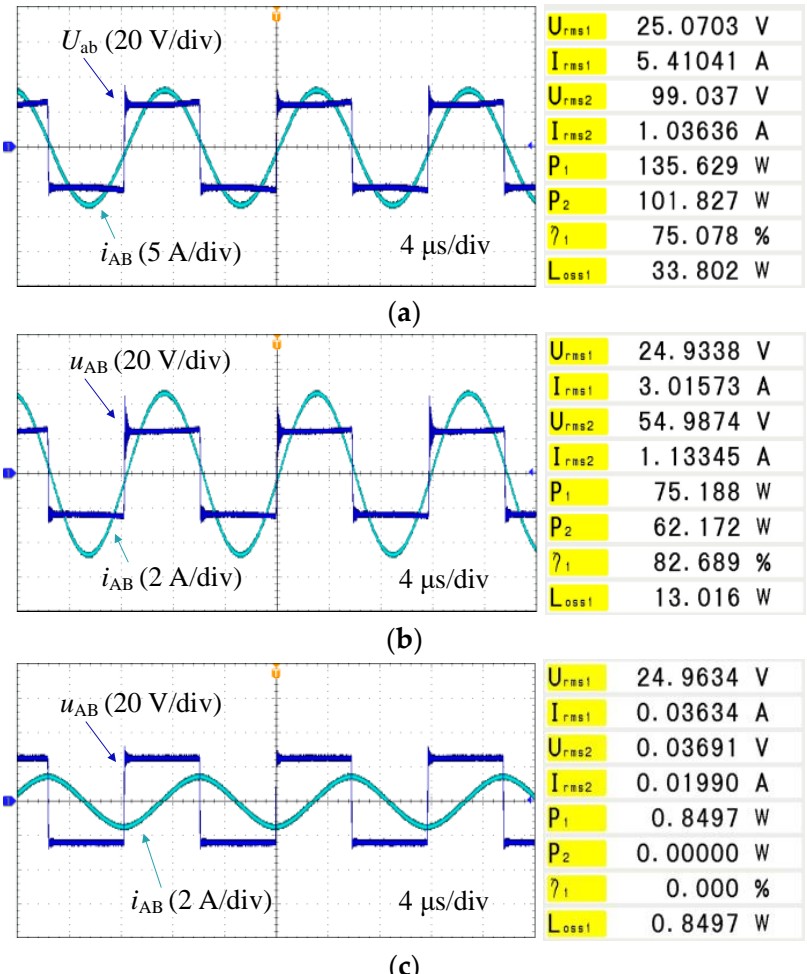

**Figure 12.** Measured waveforms and results of the IPT system with $\alpha = 3$. (**a**) $k = k_0$, $Q = 2$. (**b**) $k = k_0$, $Q = 4$. (**c**) $k = 0$.

In order to describe the characteristics of the system more clearly, the experimental results of the IPT system in the fully coupled and uncoupled state for different $\alpha$ and $Q$ are summarized in Table 2. It should be noted that the RMS value of current is used here, and the peak current can also be used to calculate *SR*. It can be seen from Figure 12a,b that when $\alpha = 3$, the transmitting coil current with $Q = 2$ is close to twice as large as that of $Q = 4$ in in the fully coupled state. However, regardless of $Q = 2$ or $Q = 4$, the transmitting coil current is the same in the uncoupled state. Therefore, the *SR* for $Q = 2$ is also about twice as large as that of $Q = 4$. In particular, when $\alpha = 3$, $Q = 4$, the no−load current of the S−LCL topology is 1.12 A, and the *SR* is approximately 3.08. The experimental results are consistent with the theoretical analysis; slight errors are caused by the measurement error or inherent deviation of coils and compensation elements, which are within the allowable range.

The electromagnetic environment problem is mainly caused by the high frequency current in the coupling coil. The high frequency electromagnetic field around the whole coupling mechanism will affect the safety of the equipment and the organisms in the adjacent area. The magnetic fields of the coupling coil and the uncoupling coil are compared using ANSYS Maxwell software. As shown in Figure 13, the simulation coil is set up as the experimental coil, consisting of 26 turns with an inner diameter of 20 cm and an outer diameter of 28 cm, to verify the flux attenuation capacity of the standby transmitting coil.

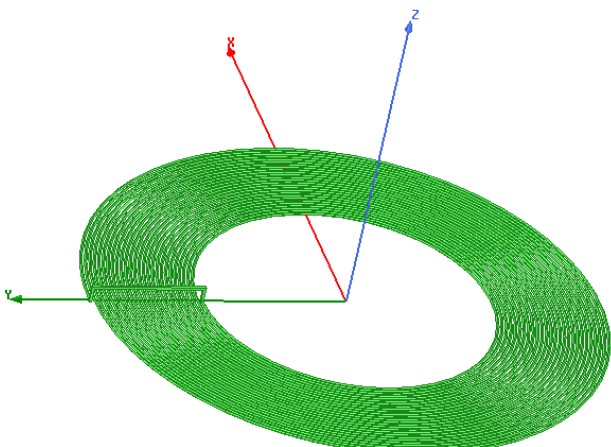

**Figure 13.** Coil setup used in the ANSYS Maxwell simulation.

According to the International Commission on Non−Ionizing Radiation Protection (ICNIRP−2010) guidelines for electromagnetic radiation, the public exposure limit for magnetic flux density is 27 µT in a range of 3 kHz–100 MHz. By substituting the current data of $\alpha = 3$, $Q = 4$ from Table 2, the electromagnetic field distribution around the system can be solved, as shown in Figure 14. The red dots indicate a magnetic flux density of 27 µT. All of the red dots join together to form a closed curve. The magnetic flux density in the inner region of the closed curve is greater than 27 µT, and the outer region is the safe zone according to ICNIRP guidelines. If the influence of the receiving coil on the magnetic field distribution is not taken into account, the magnetic field distribution diagram is shown in Figure 14a when the transmitting coil and the receiving coil are completely coupled. It can be seen that the stray field emission standard is met at a distance of 315 mm to the center of the coil in the Y−axis direction, or a distance of 396 mm to the center of the coil in the Z−axis direction. The large magnetic field ensures the power transfer capacity of the system. In practical applications, due to the magnetic flux guidance and magnetic flux leakage shielding of the receiving coil, there is basically no magnetic field leakage problem when the transmitting and receiving coils are completely coupled. The magnetic field distribution of the standby transmitting coil of the IPT system with the S−LCL topology is shown in Figure 14b; the stray field emission standard is met at a distance of 245 mm to the center of the coil in the Y−axis direction, or a distance of 235 mm to the center of the coil in the Z−axis direction.

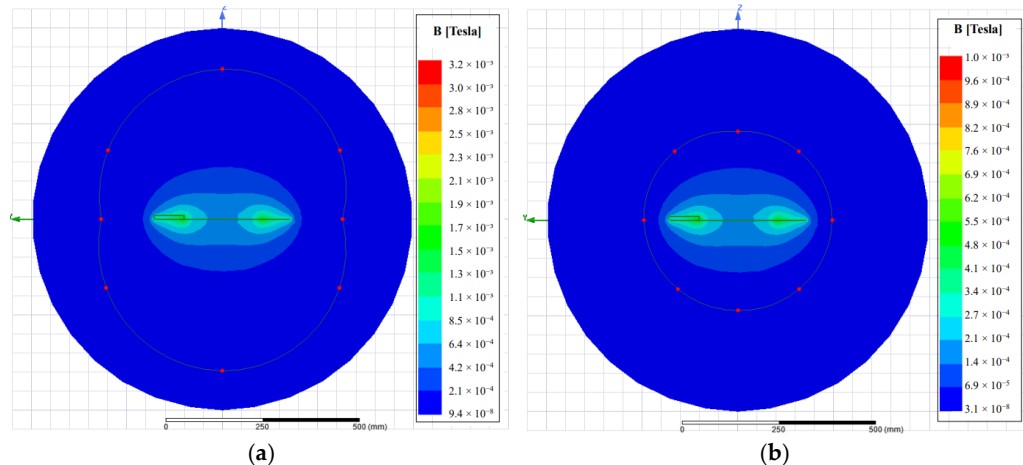

**Figure 14.** Simulated magnetic flux density of the transmitting coils. (**a**) The completely coupled state. (**b**) The uncoupled state.

In summary, when the receiving coil is coupled with the transmitting coil, the magnetic field is enhanced, so the receiving coil induction voltage is higher and the transfer power of the system is larger. When the receiving coil is decoupled from the transmitting coil, the magnetic field is weakened to ensure the safety of the surrounding electromagnetic environment without additional shielding measures. This also shows that the system designed in this paper is feasible and effective.

In order to further verify the field focusing capability of the system, transmitting coil currents with different coupling coefficients were measured, which are shown in Figure 15. Similarly, the power measurements, as well as the theoretically calculated output power from (27), are shown in Figure 16. It should be noted that there are voltage drops for various components, such as MOSFET, the capacitor and the inductor in the experiment, which are not considered in the theoretical analysis, so $U_D$ is reduced from 25 V to 23 V in the theoretical calculation. The measured results of the transmitting coil current are basically consistent with the theoretical calculation. The measured results of the power transfer are smaller than the calculated value (under high coupling coefficients). The reason for this mismatch is that the losses of power electronic devices and the losses of parasitic resistances are ignored in theoretical calculations.

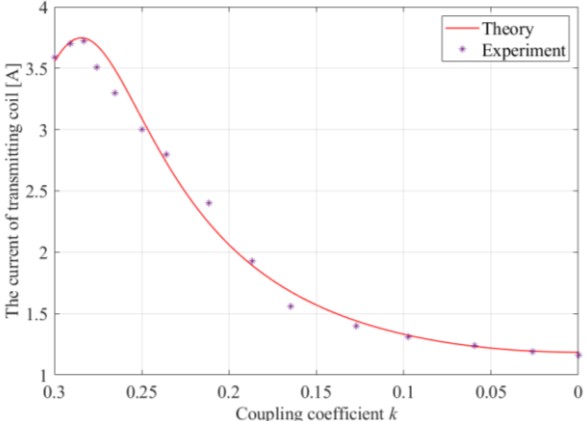

**Figure 15.** The theoretical and experimental measurements of transmitting coil currents with respect to $k$ under $\alpha = 3$, $Q = 4$.

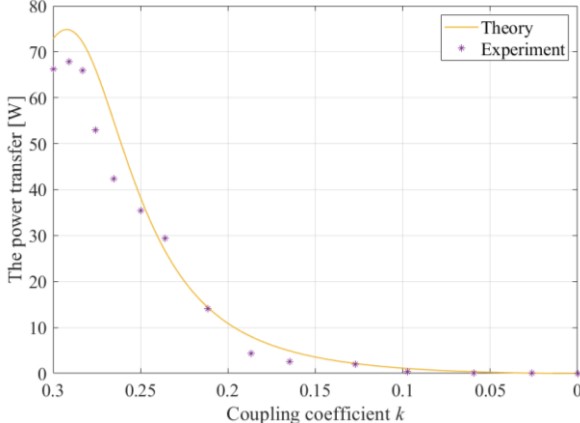

**Figure 16.** The theoretical and experimental measurements of output power with respect to $k$ under $\alpha = 3$, $Q = 4$.

The system losses and efficiency measurements with respect to the coupling coefficient $k$ are shown in Figure 17. It can be seen that the S−LCL topology can achieve high efficiency operation when the coupling coefficient is large, and the system losses also sharply decrease when the coupling coefficient becomes small. It can be seen from Figures 15–17 that the

system proposed in this paper can effectively improve the magnetic field focusing ability, and reduce the "idle" losses of the standby transmitter coils, which verifies the correctness of the theoretical analysis. However, the power transfer ability of the IPT system will be affected. If a coil or coil array with high−misalignment tolerance is used in the IPT system proposed in this paper, the power transfer capability can be greatly improved.

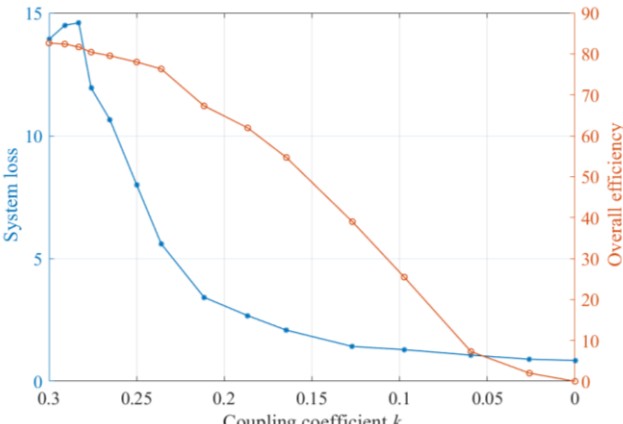

**Figure 17.** The system loss and efficiency measurements with respect to the coupling coefficient *k* under *α* = 3, *Q* = 4.

## 6. Conclusions

In order to reduce the EMF leakage and improve the overall efficiency of the IPT system, a new magnetic field containment method based on reflective properties is proposed. The experimental results show that the ratio of the coupled branch current to the uncoupled branch current (*SR*) of the IPT system is approximately 3.08 when *α* = 3 and *Q* = 4. The *SR* is about 5.97 when *α* = 3 and *Q* = 2. The IPT system with the S−LCL topology can automatically adjust the transmitting coil current without complex shielding circuits, switches, electronic devices and communications. The proposed system charges a moving load by sending pulses of power through segmentation transmitting coils. Power flow regulation can be controlled by adjusting the number of pulses. It is worth noting that the proposed method in this paper can be used in conjunction with other proposed methods. If a coil or coil array with high misalignment tolerance is used in the IPT system with S−LCL topology, the anti−misalignment capability and power transfer capability of the system can be greatly improved, which is the research direction of the authors.

**Author Contributions:** Conceptualization, J.Y.; methodology, X.Y.; software, J.F.; validation, X.Y. and J.Y.; formal analysis, B.W.; investigation, D.L.; data curation, J.Y.; writing—original draft preparation, X.Y.; writing—review and editing, X.Y. and J.Y.; visualization, J.F.; supervision, J.Y.; project administration, X.Y.; funding acquisition, B.W. All authors have read and agreed to the published version of the manuscript.

**Funding:** This work was supported in part by the National Natural Science Foundation of China under Grant 12004202.

**Conflicts of Interest:** The authors declare no conflict of interest.

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
