# Peer review of "A Magnetic Field Containment Method for an IPT System with Multiple Transmitting Coils Based on Reflective Properties"

_electronics, doi:10.3390/electronics12030653_

Round 1

Reviewer 1 Report

Inductive power transfer was first studied by Tesla and it is based on the magnetic field. An important field should be conserved for an efficient transmission. In inductive power transfer is important the magnetic field, not generally electromagnetic field as you mentioned in abstract. Charging of mobile phones based on the inductive coupling are already well-known. The relations including impedance should be corrected because they are also complex values (4-9 and 11-18). For the proposed model you must specify also what are the limits for the transmitted power. It is not clear in what consist the reflective properties, for what? You do not have a comparison for usual materials in such case. Also did you take some measure for human safety if the values of the magnetic field are important?

Author Response

Dear editor and reviews,

I would like to thank the respected reviews for their constructive comments on our manuscript “A Magnetic Field Containment Method for IPT System with Multiple Transmitting Coils Based on Reflective Properties (Electronics-2115949)”. I have considered the comments very carefully and have revised the paper accordingly. All changes to the text and figures are shown. Thanks to the reviews, I believe that this revised paper has been improved considerably. I hope that the corrections are satisfactory.

Note: The reply comments are in the attachment.

Best regards,

Xu Yang et al.

Reviewer 2 Report

A lumped-parameter approach is employed to examine the performance of arrays of multiple transmitting coils, that are suitable for wireless electrical charging. As a measure of device performance, power transfer and electric current are examined. Experimental proofs are performed by simpler set-ups. Numerical simulations are carried out for a key device component of coupling coils. Besides, ways of reducing idle losses are discussed.

There are some typo errors and questions as follows.

(on Line 20) ‘The experiment results show’ -> ‘The experimental results show’

On Line 34) ‘multiple transmitting coils array’ ->

[i] ‘arrays of multiple transmitting coils’

[i] ‘multiple-transmitting-coil array’

(You may choose a proper one)

(a question about Figs. 1 and 2) How much difference in the usual total power flow, say, in Watt-Hour, necessary for charging a mobile phone and an electric vehicle.

The reason is that although the principle of wireless charging might the same for both applications, the attendant engineering challenges will be quite distinct. Please expand on those differences from engineering points of view.

(On line 244) ‘, The transmitting coil’ -> ‘, the transmitting coil’ 

(a question) How do the authors come up with the various device topologies (S-L, LCC-P, S-LCS, etc.)? Are there other choices for device topologies? If there are other device topologies, why are they omitted (On what grounds)?

(a question) How does one alter the coupling coefficient (k) in terms of real device manipulations in drawing Figs. 15, 16, and 17?

Author Response

(The authors gave the same response as above.)

Round 2

Reviewer 1 Report

Even you made some improvements in the manuscript, the presentation has not the necessary impact. Even I gave previously information about the necessary corrections in the relations there are still errors. You do not understand the concept of complex value. Also, the experiments do not show clearly the values for the power that can be transmitted. You mentioned that the method could support also medium powers. I doubt about that. In my opinion the manuscript must be reconsidered because only modifying some words or introducing a text is not enough.

Author Response

Dear editor and reviews,

I would like to thank the respected reviews for their constructive comments on our manuscript “A Magnetic Field Containment Method for IPT System with Multiple Transmitting Coils Based on Reflective Properties (Electronics-2115949)”. I have considered the comments very carefully and have revised the paper accordingly. All changes to the text and figures are shown. Thanks to the reviews, I believe that this revised paper has been improved considerably. I hope that the corrections are satisfactory.

Note: The reply comments are in the attached file.

Best regards,

Xu Yang et al.

Reviewer 2 Report

Many of the concerns raised by this reviewer have been resolved.

Author Response

(The authors gave the same response as above.)
